# Effect of Metakaolin on the Microstructural and Chloride Ion Transport Properties of Concrete in Ocean Wave Splashing Zones

**DOI:** 10.3390/ma16010007

**Published:** 2022-12-20

**Authors:** Yezhen Yuan, Kaimin Niu, Bo Tian, Lihui Li, Jianrui Ji, Yunxia Feng

**Affiliations:** 1School of Civil Engineering, Chongqing Jiaotong University, Chongqing 400074, China; 2Institute of Highway, Ministry of Transport, Beijing 100088, China

**Keywords:** carbonation, sulfate, three-factor coupled, pore structure

## Abstract

In order to address the problem of the durability deficiency of concrete in wave splash zones in a harsh marine environment, this paper investigates the effects of coupled carbonation, sulfate, and chloride salts on the strength, capillary water absorption, and ion migration properties of cement concrete incorporated with metakaolin, and characterizes the pore structural changes with the mercury-pressure method and AC impedance technique. The results show that, compared with a single chloride salt environment, the improvement in mortar strength and impermeability with carbonation coupling is almost positively correlated with the calcium content in the specimen, and renders its pore structure more refined and denser. In contrast, the presence of sulfate reduces mortar strength and increases the ion migration coefficient. When the three factors of sulfate, carbonation, and chloride salt were coupled, damage to the strength and pore structure of the specimens was the most significant, but the specimen incorporated with 30% metakaolin had its strength improved compared with the blank group specimen; from the perspective of pore structural parameters and transport coefficient, the microstructure was denser, and the impermeability was significantly improved.

## 1. Introduction

As the most widely used type of building structure, concrete structures play an important role in the development and utilization of marine resources. However, marine concrete buildings are facing huge challenges, among which the corrosion of reinforcement caused by chloride ion intrusion is considered to be the most important cause of damage to marine concrete [1]. Chloride ion intrusion into concrete can occur in two ways: internal admixture and external penetration [2]. Internal admixture occurs when chloride ions in the raw material are mixed into the concrete with other admixtures during the construction process. External penetration occurs when chloride ions from the external environment diffuse into the concrete in engineering. Factors affecting the permeability of cement-based materials are shown in Table 1 [3]. Permeation, diffusion, capillary adsorption, and electrochemical migration are the main ways through which chloride ions from the external environment invade the concrete [4,5]. Chloride ions invade the concrete and adsorb onto the surface of the reinforcement, decreasing the pH, which destabilizes the passivation film, thus corroding the reinforcement.

In an oceanic environment, the damage degrees of marine concrete structures in different locations are ranked as: wave splash zone > atmospheric zone > underwater zone [10]. In particular, concrete in a wave splash zone often simultaneously suffers from the combined effects of multiple factors such as carbonation, ion erosion, and dry–wet cycles. Most of the current studies on the chloride ion diffusion performance of concrete have focused on single-factor effects or sulfate-chloride salt coupling [11] and chloride-carbonation coupling [12] effects. According to the available studies, chloride ions intruding cement concrete are mainly present in the form of free chloride ions in the pore solution and combined chloride ions of reaction products. The free chloride ion concentration is related to the diffusion of chloride ions to the interface between cement and reinforcement, while the binding of chloride ions can reduce the free chloride ion concentration and inhibit chloride ion penetration into the matrix [13,14]. Sulfate ion penetration reacts with the Ca and Al phases in cement to form expansion products such as calcium alumina and gypsum. The formation of calcium alumina leads to an increase in crystallization pressure, which, in turn, causes the expansion and cracking of concrete. The cracking of concrete, in turn, causes the further intrusion of sulfate ions and other aggressive ions [15,16]. In Portland cement, CO_2_ dissolved into the pore solution reacts with Ca(OH)_2_ and C-S-H gels produced by the hydration of Portland cement to form CaCO_3_ [17], decreasing the pH and increasing the risk of the corrosion of the reinforcement, leading to a reduced normal service life of the structure. Liu et al. [18] found that carbonation reduces the chloride ion binding capacity and allows for chloride ions to diffuse into the concrete interior. Sulfate ions first react with AFm in hardened cement paste to form calcium alumina and destabilize the existing Friedel’s salt to decompose it, which weakens the concrete’s ability to solidify chloride ions and promote their transport within the concrete [19,20]. Resistance to chloride ion penetration can be improved by using new cementitious materials [21] and recycled coarse aggregates [22]. There are many studies showing that, for cementitious materials, the addition of mineral admixtures can effectively improve their resistance to chloride ion attack [23]. Compared to common admixtures such as fly ash, metakaolin is more effective in improving the performance of concrete against chloride ion penetration and sulfate attacks [24]. This is mainly because the incorporation of metakaolin can refine the pore structure of concrete [25]: a large number of pores are filled to reduce the capillary content, thus reducing the chloride ion transport channels. Moreover, the greater the incorporation amount is, better the refinement effect. In addition, metakaolin can improve the stability of bound chlorides such as Friedel’s salts.

Research on the influence of three-factor coupling on chloride ion transport under a dry–wet cycle is still lacking. The marine environment is complex and changeable, so it is necessary to simulate a real marine environment as accurately as possible under laboratory conditions. This paper investigates the chloride ion transport properties of cement mortar specimens doped with metakaolin and fly ash under the coupling of carbonation, chloride salt and sulfate combined with MIP and AC impedance spectroscopy to characterize the microstructural changes and to provide a reference for improving the chloride ion penetration resistance of concrete within a complex marine environment.

## 2. Materials and Methods

### 2.1. Raw Materials and Specimen Preparation

#### 2.1.1. Raw Materials

The cement was P.O.42.5 Ordinary Portland Cement (GB175-2007) produced by Hebei Dingxin Cement Co., Ltd. (Baoding, China) Metakaolin produced by the Shijiazhuang Lingshou Zhongshan Cement Group, and low-calcium fly ash produced by Wuhan Weishen Technology Co., Ltd. (Wuhan, China) were used as mineral admixtures, and the chemical composition analysis is shown in Figure 1. The water reducing agent was the polycarboxylic acid system high-efficiency water reducing agent produced by Guangzhou Sihang Material Technology Co., Ltd. (Guangzhou, China). The fine aggregate was river sand with a fineness modulus of 2.65, an apparent density of 2690 kg/m^3^, and a mud content at 1.0%. The coarse aggregate was 5–10 mm continuous graded gravel with an apparent density of 2810 kg/m^3^.

#### 2.1.2. Specimen Preparation

The mortar and concrete ratios prepared for the test are shown in Table 2. Using 30% supplementary cementitious materials (SCMs) instead of cement is beneficial in enhancing the impermeability of concrete [26,27]. The mortar specimen size for the compressive strength and capillary water absorption test was a 40 mm × 40 mm × 40 mm cube, and the concrete specimen size for the rapid chloride migration test was a Φ100 × 50 mm cylinder. Molding in a standard curing room (temperature, 20 ± 2 °C; relative humidity, over 95%) for 28 days was followed by a corrosion test.

#### 2.1.3. Corrosion Regime

The four different corrosion regimes for oceanic environments were set up in this study as follows.

(a) Single chloride salt environment (Cl): the specimen was first immersed in a concentration of 5% NaCl solution for 48 h, followed by drying at 60 °C for 6 h, and then placed in a room-temperature environment for 18 h to complete a dry–wet cycle.

(b) Carbon–chlorine salt coupled environment (CCl): the specimen was first immersed in a 5% NaCl solution for 48 h, followed by drying at 60 °C for 6 h, and then placed in an environment with a CO_2_ concentration of (20 ± 3)%, temperature of (20 ± 5) °C and relative humidity of (70 ± 5)% for 18 h to complete a coupling cycle.

(c) Sulfate–chlorine salt coupling environment (SCl): the specimen was first immersed in 5% NaCl +10% Na_2_SO_4_ solution for 48 h, followed by drying at 60 °C for 6 h, and then placed in a room-temperature environment for 18 h to complete a coupling cycle.

(d) Sulfate–carbonation–chlorine salt coupled environment (SCCl): the specimen was first immersed in a concentration of 5% NaCl + 10% Na_2_SO_4_ solution for 48 h, followed by drying at 60 °C for 6 h, and then placed in a CO_2_ concentration of (20 ± 3)%, temperature of (20 ± 5) °C, relative humidity of (70 ± 5)% environment for 18 h to complete a coupling cycle.

### 2.2. Test Methods

#### 2.2.1. Compressive Strength Test

The compressive strength of the mortar specimens before corrosion and after 32 cycles of corrosion was determined according to the GB/T 17671-2021 testing method for cement mortar strength (ISO method) using an automatic pressure tester with a constant loading rate of 300 N/s. The average value was obtained by testing three specimens each time.

#### 2.2.2. Capillary Water Absorption Test

The capillary water absorption test was performed on the mortar specimens before corrosion, and after 10 and 20 cycles of corrosion. The specimens were first dried to a constant weight in an oven at 105 °C; then, all surfaces except for the test surface were sealed with paraffin and placed onto a tank stand at a temperature of 20 ± 1 °C, with the test surface facing downward to ensure one-way water transfer; see Figure 2. During the test, the water level was maintained at 3 mm above the bottom of the specimen, and the specimen was taken out every 15 min. The specimens were weighed immediately with a wet cloth to clear the surface water. The capillary water absorption coefficient could be obtained from the relationship between the total amount of capillary water absorption and the square root of time, as shown in Equation (1).
(1)$$\frac{M(t)}{A}=S\cdot t^{0.5}$$
where M(t) is the amount of absorbed water, kg; A is the cross-section of the specimen in contact with water, m^2^; S is the capillary absorption coefficient, kg/(m^2^∙h^0.5^); t is the time, h.

#### 2.2.3. Rapid Iodine Ion Migration Test

The rapid chloride migration method (RCM) is widely used to assess the chloride permeability of concrete, but it is not applicable to concrete that has been attacked by chloride ions because chloride ions existing in the concrete migrate under electrically accelerated conditions and react with the silver nitrate chromogenic agent, affecting the accuracy of the RCM test results. For this reason, Lay S [28] used iodine ions instead of chloride ions as the cathodic solution, and potassium iodate-acetic acid-starch as the color developer, i.e., the rapid iodine ion migration test (RIM). The ion diffusion coefficient of concrete could be obtained from Equation (2) [29].
(2)$$D=\frac{0.0239\times (273+T)L}{(U-2)t}(X_{d}-0.0238\sqrt{\frac{(273+T)LX_{d}}{U-2}})$$
where D is the ionic diffusion coefficient of concrete (×10^−12^ m^2^/s); U is the absolute value of voltage (V); T is the average temperature of the anode solution (°C); L is the thickness of the specimen (mm); X_d_ is the average ionic penetration depth (mm); t is the duration of the test (h).

The test setup for the RIM test is shown in Figure 3, including a 0.3 mol/L NaOH solution and a 11.89 wt.% NaI solution; lastly, the diffusion depth of the iodine ions in the concrete was measured, as shown in Figure 4. The migrations of iodide ions and chloride ions were very similar during the test, and there was a strong linear relationship between the diffusion coefficient of iodide ions (DRIM) and the diffusion coefficient of chloride ions (DRCM).

#### 2.2.4. Mercury Injection Test

The pore structure of each specimen group after corrosion was determined using an AutoPore IV 9500 mercury-pressure instrument from Micromeritics, USA. The pressure range was 0.2–50 psia in the low-pressure chamber, and 0–60,000 psia in the high-pressure chamber, and the pore size could be measured in the range of 1080–0.003 μm.

#### 2.2.5. Alternating-Current Impedance

The specimens before and after corrosion were measured by using a HIOKI IM3570 impedance analyzer, as shown in Figure 5. And the test frequency range was from 5 MHz to 1000 Hz. The amplitude of the sinusoidal voltage was 10 mV, and the Nyquist curve was obtained by fitting the test data with software ZView 3.1. The specimens had to be treated with vacuum water retention before the test, and the filter paper had to be soaked in a NaOH solution and placed between the specimen and the electrode with a clamp to ensure close contact between the filter paper and the specimen. The specimen size was 40 mm × 40 mm × 40 mm.

## 3. Results and Discussion

### 3.1. Compressive Strength

The compressive strength was determined for each group of mortar specimens in standard curing for 28 days and for mortar specimens after further exposure to single chlorine salt, carbonate-chlorine salt, sulfate-chlorine salt double-factor coupled, and sulfate-carbon-chlorine salt triple-factor coupled erosion environments for 20 cycles after 28 days of standard curing; the results are shown in Figure 6. The initial strength was 49.7 MPa for Specimen M mixed with 30% metakaolin, which was about 3 MPa higher than that of Specimen C without mineral admixture, followed by 47.2 MPa for Specimen MF mixed with a 1:1 ratio of metakaolin and fly ash, while the initial strength of Specimen F mixed with 30% fly ash was about 3 MPa lower than that of C. This shows that metakaolin had a certain enhancement effect on the cement mortar, which is consistent with the study by Sujjavanich et al. [26]. After single chlorine salt erosion, the strength of the mortar specimens in each group did not decrease significantly and even slightly increased, which was due to the filling effect of F salt on the pores. When subjected to coupled carbonation-chlorine salt erosion, the strength of all groups of mortars increased, with that of Specimen C increasing the most by 5.64%, followed by F and MF by 3.93% and 2.88%, respectively, while M was the lowest at 2.35%, which was positively correlated with the calcium content in the specimen, indicating that the generation of carbonation product CaCO_3_ rendered the matrix structure denser, leading to an increase in strength. The strength of the mortars exposed to chloride-sulfate coupling showed a decreasing trend for all groups, and the phenomenon was exacerbated by the further coupling of carbonation, where the most obvious strength loss was in Specimen C without admixture, whose strength loss was 13.4% under the three-factor coupling, indicating that the mineral admixture, in particular the metakaolin, could improve its resistance to erosion to some extent. The destructive effect of sulfate on cement mortar was stronger, probably because the generated gypsum, calcium alumina, and other expansive substances destroyed the internal structure of the material, and the coupled effect of carbonation further accelerated the decomposition of the cement hydration products by decalcification.

### 3.2. Capillary Water Absorption Performance

The evolution of water absorption per the unit area of different cement mortar specimens after 10 and 20 cycles of exposure to different erosive environments is shown in Figure 7. The water absorption per unit area increased with the square root of time for each group of mortars and could be roughly divided into two stages. In the first stage, the water absorption per unit area increased rapidly in a linear relationship with the square root of time; after that, due to the gradual saturation in the mortar interior, the capillary water absorption weakened. In the second stage, the water absorption curve gradually leveled off, and the water absorption rate decreased, but the water absorption per unit area still showed a linear relationship with the square root of time. By linearly fitting the water absorption per unit area with the square root of time, the fitting correlation coefficient was R^2^ > 0.9, and the slope of the fitted equation in the first stage was defined as the capillary absorption coefficient [30]; the results are listed in Table 3.

Figure 7a shows that the capillary water absorption and water absorption coefficient of each group of the mortar specimens mixed with a mineral admixture were lower than those of blank cement mortar Specimen C. This was because the admixture of metakaolin could significantly refine the mortar pore size and reduce the water absorption. When the specimens were placed in a single chlorine salt environment, the water absorption and water absorption rate gradually decreased with the increase in corrosion cycles. After 20 cycles, the early capillary absorption coefficient of each group of mortar gradually tended to be consistent and was maintained at 0.6–0.7 g/(cm^2^∙min^0.5^), among which the C and F groups were significantly reduced compared to 10 cycles. The M and MF groups changed little because of, on the one hand, a large number of F salt filling the pores, so that the mortar structure with a greater capillary pore content was denser, preventing water transfer. On the other hand, the M and MF specimens had a denser structure themselves, so the filling of a large amount of F salts may shrink the larger pores into small pores that are conducive to capillary action to some extent rather than increasing the capillary water absorption. When corroded in the coupled carbonation–chlorine salt environment for 10 cycles, the capillary water absorption of each group of the mortars was significantly higher than that of the single chlorine salt environment except for Specimen C. The capillary water absorption of Specimen C was reduced to the lowest after 20 cycles of corrosion, indicating that the mortar specimens without an admixture were the most significantly affected by carbonation, and the large amount of generated CaCO_3_ filled the pores and enabled the matrix to become denser. The capillary water absorption of the other groups with an admixture decreased in the order of F, M, and MK; thus, it is speculated that the carbonation products mainly have an obvious filling effect on the large pores. When corroded in the chloride–sulfate coupled environment for 10 cycles, the capillary water absorption coefficient of each mortar group was significantly increased compared with the single chloride-sulfate environment, except for Specimen M. This may have been due to the formation of a large amount of calcium alumina, resulting in the increase in microcracks within the cement matrix, thus increasing the early capillary water absorption. After 20 cycles, due to the further formation of erosion products such as calcium alumina and F salts, the filling effect of the pore structure was greater than the destruction effect of the capillary pores, so capillary water absorption was reduced. However, the capillary water absorption rate in the second stage was smaller than that in other environments, indicating that the effect of sulfate on mortar pore structure was stronger. When corroded under the coupling of three factors for 10 cycles, the capillary water absorption of each group of the mortar specimens increased linearly with time, and the capillary water absorption coefficients of the two stages were closer; especially in the M and MF groups, the capillary water absorption coefficient was stable at about 0.65 g/(cm^2^∙min^0.5^). This indicates that the filling effect of erosion products on the pores was more obvious when sulfate and carbonation were present at the same time, with sulfate corrosion products accounting for a larger proportion.

### 3.3. Ion Migration Performance

The migration performance of the iodine ions in each group of concrete before and after corrosion is shown in Figure 8. The iodine ion migration coefficient (DRIM) was the smallest for Specimen M after corrosion, followed by those of MF and F. This indicates that the addition of admixtures could significantly improve the pore structure of concrete; the specimen with 30% metakaolin had the greatest improvement in resistance to iodine ion penetration. Sujjavanich et al. [26] found a similar regulation. When exposed to both single chloride salt and coupled carbonation-chloride salt environments, the DRIM decreased to some extent as the number of corrosion cycles increased. A single chlorine salt attack generally does not cause damage to the concrete structure; instead, chlorine salt crystallization under dry–wet cycle conditions refines the pore size. After the coupling of carbonation–chlorine salt, carbonation products grew in the pore space, further blocking the pore size and improving its impermeability, so the specimens had the smallest DRIM in this environment. On the other hand, when exposed to the sulfate-chlorine salt and sulfate-carbonation-chlorine salt coupled environments, the DRIM of each group of the concrete specimens increased to some extent with the increase in the number of corrosion cycles, especially when the three coupled factors increased more significantly. This indicates that the existence of sulfate destroyed the pore structure of cement due to the generation of expansion products such as gypsum and calcium alum, and accelerated the penetration of corrosive agents. Carbonation accelerated the neutralization process of concrete, thus causing the unstable decomposition of hydration products such as Ca(OH)_2_ and C-S-H gels, and further accelerating the structural damage. The degree of DRIM decrease was related to the specimen Ca content after the carbide–chlorine salt coupling in the order of C, F, MF and M, which shows that the filling of more carbonation products in the pores could significantly reduce the transport of external corrosion agents. However, the DRIM growth of Specimen C was also the largest for the concrete exposed to the coupled sulfate–carbonation–chloride salt environment. Overall, mineral admixtures, especially metakaolin admixtures, greatly enhance the impermeability of concrete in oceanic wave splash zones.

### 3.4. Analysis of the Pore Structure

In order to clarify more the mechanism of the effect of metakaolin on the chloride transport properties of concrete within an oceanic environment, the pore structure was investigated for different mortar specimens under the three-factor coupled environment and for Specimen M with 30% metakaolin in different chloride salt environments, and the total pore volume and the pore volume in different size ranges were measured; the results are shown in Figure 9. When Specimen M was corroded in different chlorine salt environment after 20 cycles, the coupled sulfate–carbonation–chlorine salt environment caused the most serious damage to its pore structure. The total pore volume content of the M-SCCl specimen was the largest, followed by that of M-SCl, and both 50–1000 and >1000 nm large pore contents were also significantly higher than those of the other two groups of specimens. In contrast, M-CCl had the lowest total pore volume and macropore content, and the highest gel pore (<20 nm) content, indicating that carbonation refined the pore size, which led to the lowest DRIM for M-CCl. However, for different concrete specimens exposed to the three-factor coupled environment, M-SCCl still had the lowest total pore volume content, and the gel pore content of the three groups of specimens blended with mineral admixtures was significantly greater than that of pure cement specimen C-SCCl, particularly MF-SCCl, with a mixture of metakaolin and fly ash.

The fractal dimension model of concrete pore volume can be used in the Menger sponge configuration and combined with mercury intrusion data to obtain the fractal dimension of pore volume. The larger the fractal dimension of pore volume is, the lower the porosity of concrete, the increase in pore surface area, and the better pore structure are [31]. The expression for the fractal dimension solution is:(3)lg(−dν/dr)∞(2−V)lgr
where v is the pore volume; r is the pore diameter; V is the fractal dimension of the pore volume.

The fitting curves of the pore volume fractal dimension calculated for each group of specimens in the three-factor coupled environment are shown in Figure 10; the R^2^ of each group was greater than 0.9, indicating that it had good fitting correlation. The pore volume fractal dimensions of M-SCCl, MF-SCCl and F-SCCl were 3.542, 3.465, and 3.396, respectively; the pore volume fractal dimension of M-SCCl was obviously the largest, followed by those of MF-SCCl and F-SCCl. It further confirmed that the incorporation of metakaolin could improve the pore structure of cement mortar under the coupling effect of three factors. Furthermore, the best performance in this environment was that of the single admixture of metakaolin.

### 3.5. Electrochemical Impedance Analysis

Impedance spectroscopy is widely used in the nondestructive characterization of the dielectric properties, internal microstructure, and cement hydration process of cement-based materials. The microstructure of concrete is illustrated by the simplified model shown in Figure 11a, in which there are three paths to conduct alternating currents: the continuous conduction path (CCP), the discontinuous conduction path (DCP) and the “insulated” conduction path (ICP) [22]. The continuous conduction path (CCP) consists of a series of connected capillary pore channels in which the pore solution plays a major role in transmission. The discontinuous conduction path (DCP) has pore channels separated by particles of gelling material or its hydration products. The “insulating” conduction path (ICP) consists of aggregate and particles of gelling material or hydration products. The “insulating” conduction path (ICP) consists of the aggregate and the particles of the gelling material or hydration products. The continuous conduction path in the concrete microstructure can be represented by a resistor (RCCP), a resistor (RCP), and a capacitor (CDP) connected in series to represent its discontinuous conduction path, and a capacitor (CICP) to represent its “insulated” conduction path. Then, the electrical components representing the three paths are connected in parallel to obtain the corresponding equivalent circuit as shown in Figure 11b, and it is further simplified to the equivalent circuit shown in Figure 11c, which corresponds to the Nyquist diagram shown in Figure 12 [32]. The equivalent circuit shows that R0 corresponds to the intersection of the left-hand side of the semicircle with the real axis in the Nyquist diagram, and R1 corresponds to the diameter of the semicircle in the Nyquist diagram.

In the equivalent circuit model proposed by Song [32], the resistance of all continuously and discontinuously connected micropores that were filled with water in concrete can be described as R_0_, which was inversely proportional to the porosity. The reaction kinetic parameter R_1_ could be obtained from the diameter of the high-frequency semicircle. R_1_ is the kinetic parameter of concrete for hydration reaction or other chemical reactions, and is a sign of the degree of hydration of concrete. R_0_+R_1_ represents the electrical resistance of all connected pore solutions in concrete, i.e., the R_CCP_ parameter could characterize the connected porosity of concrete.
(4)$$R_{0}=R_{CP}R_{CCP}/\left(R_{\mathrm{CP}}+R_{CCP}\right)$$
(5)$$R_{1}=R_{\mathrm{CCP}}^{2}/\left(R_{CP}+R_{CCP}\right)$$
(6)$$C_{1}={\left(1+R_{CP}/R_{CCP}\right)}^{2}C_{\mathrm{DP}}$$
(7)$$R_{\mathrm{CCP}}=R_{0}+R_{1}$$

The Nyquist plots of each group of specimens under different erosion environments are shown in Figure 13. The values of impedance parameters R_0_ and R_1_ of the specimens were obtained by fitting the Nyquist curves with Zview software, the RCCP parameters were further calculated, and the results are listed in Table 4. The high-frequency arc diameters of Specimens M and MF were significantly larger than those of the other two groups before corrosion, indicating that the incorporation of metakaolin had a significant effect on refining the pore structure and improving the density of the substrate. After 32 cycles of corrosion under the single chloride salt and coupled carbonization-chloride salt environments, the impedance parameters of each specimen increased, especially for the specimens with mineral admixture. Among them, Specimens MF and F had a greater increase in electrical resistance after carbonization compared with that of M. This was related to the higher calcium content in both, but Specimen C, with the highest calcium content, changed little after carbonization, which may have been related to its own larger pores before carbonization. When the specimens were exposed to the coupled erosion of sulfate-chlorine salt and sulfate-carbonation-chlorine salt, the impedance parameters of the specimens were all reduced to different degrees compared to the single chlorine salt erosion, among which, for Specimen C, the impedance parameters under the sulfate–chloride salt coupling (SCl) were smaller than those of the three factors (SCCl) together, indicating that the coupling of carbonation could reduce the damage of the matrix pore structure by sulfate to a certain extent. For the pure cement mortar specimen with more pores, the filling of carbonation products could reduce the entry of sulfate to some extent. However, for Specimen M, the impedance parameter in the SCl environment was significantly larger than that in the SCCl environment, indicating that, for the mortar specimen incorporated with metakaolin, the decomposition of hydration products was more significantly accelerated by the neutralization due to carbonation, thereby deteriorating the pore structure. Comprehensively, the R_CCP_ value of cement mortar Specimen MF, prepared by compounding metakaolin and fly ash, was the largest at 2976 Ω, followed by M at 2031 Ω, and the lowest at 1256 Ω for Specimen C, after 32 cycles of corrosion under the coupling effect of the three factors. This indicates that the incorporation of metakaolin could greatly improve the erosion resistance of cement concrete under the coupling effect of multiple factors in an oceanic environment. This is consistent with the results of the previous study.

## 4. Conclusions

(1) Compared to the single chloride salt environment, the coupling of carbonation increased the mortar strength of all groups of specimens and was positively correlated with the calcium content in the specimens, but the coupling of sulfate decreased the mortar strength, and the three-factor coupling aggravated this phenomenon, especially for pure cement mortar with a strength loss of 13.4%. However, the compressive mortar strength of the specimens incorporated with 30% metakaolin increased compared to the blank group, whether exposed to a single chloride salt environment or to a multifactor coupling environment.

(2) Compared with the blank group, when the 30% metakaolin was incorporated, the capillary water absorption coefficient of the concrete specimens was significantly lower in both the single chloride salt attack environment and the multifactor coupled attack environment. The iodine ion migration coefficient was the smallest after 32 cycles of corrosion under different attack environments. The migration coefficient of each group of concrete specimens was the largest after 32 cycles of corrosion in the coupled carbide-sulfate-chlorine salt environment, while the migration coefficient was the smallest after 32 cycles of corrosion in the coupled carbide-chlorine salt environment, indicating that the carbonation products filled its pores, thus improving its permeability resistance.

(3) The sulphate–carbon–chloride three-factor coupling caused the most severe damage to the cement stone pore structure, followed by the sulphate–chloride coupling, whereas the carbon–chloride coupling refined its pore size instead. When exposed to the three-factor coupling environment, the specimen incorporated with 30% metakaolin had the densest pore volume and better impermeability. This indicates that metakaolin significantly improved the microstructure of concrete in the wave splash zone within a severe oceanic environment.

## Figures and Tables

**Figure 1 materials-16-00007-f001:**
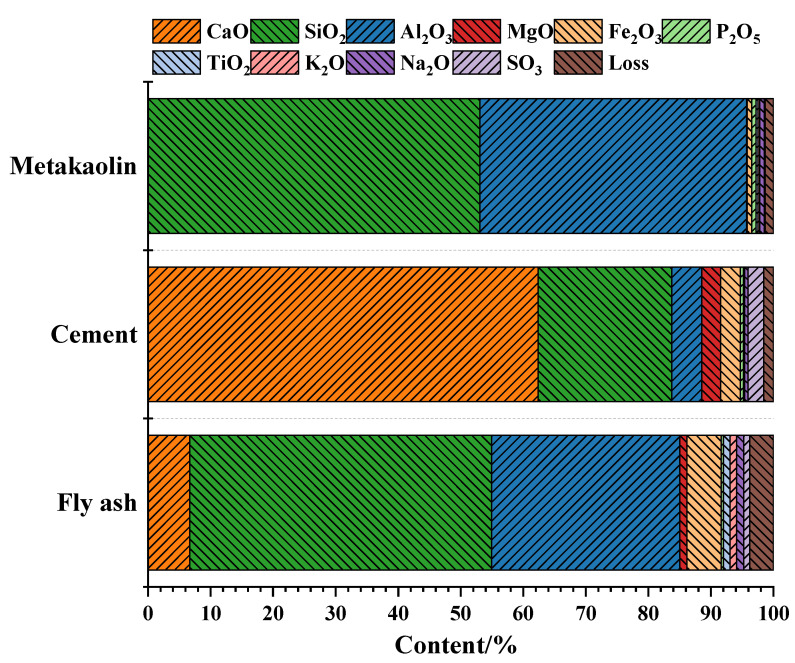
Chemical composition analysis of raw materials.

**Figure 2 materials-16-00007-f002:**
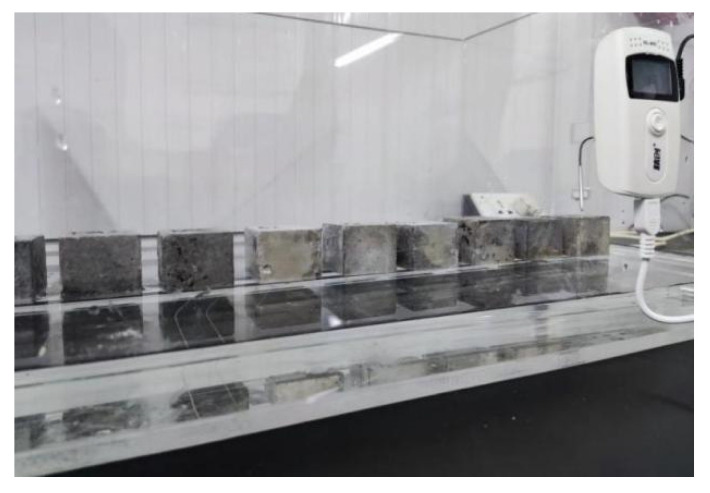
Capillary water absorption test device figure.

**Figure 3 materials-16-00007-f003:**
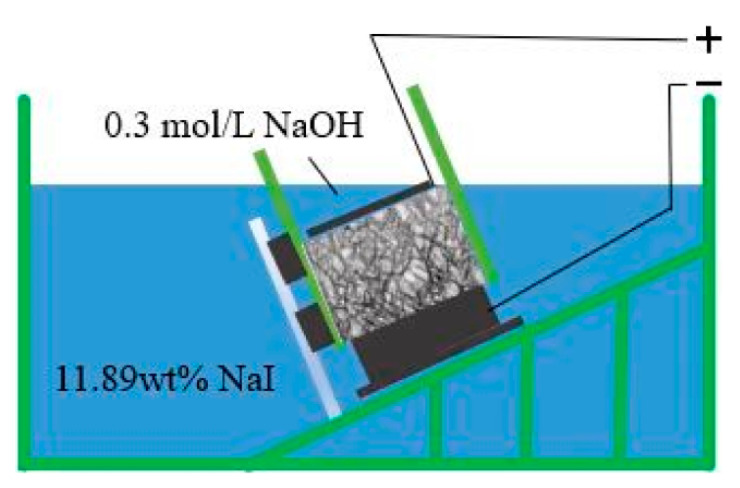
RIM test setup.

**Figure 4 materials-16-00007-f004:**
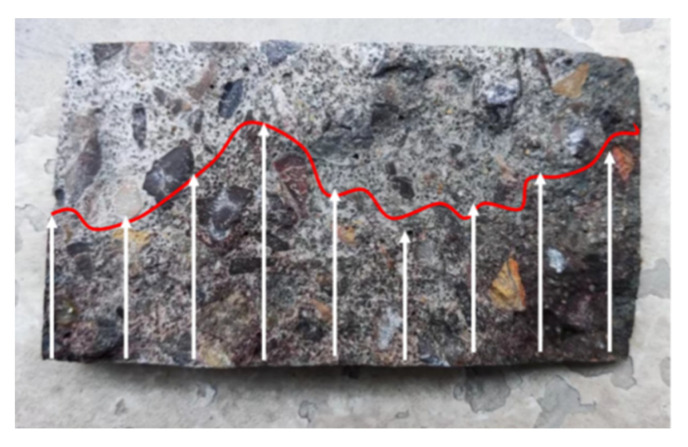
Penetration depth of iodine ions.

**Figure 5 materials-16-00007-f005:**
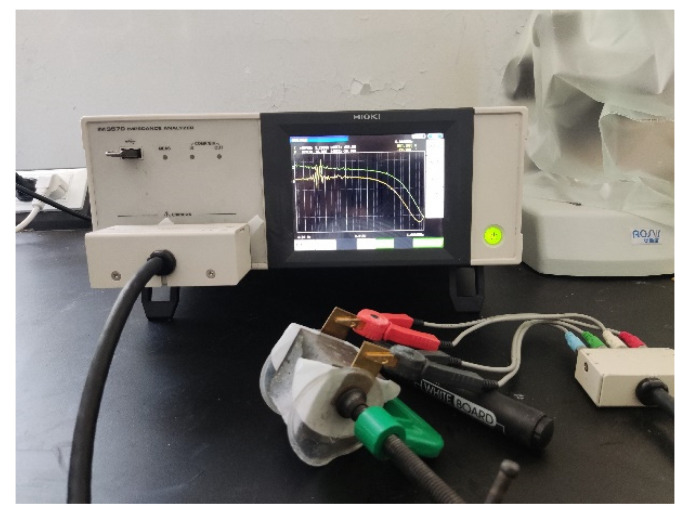
AC impedance test.

**Figure 6 materials-16-00007-f006:**
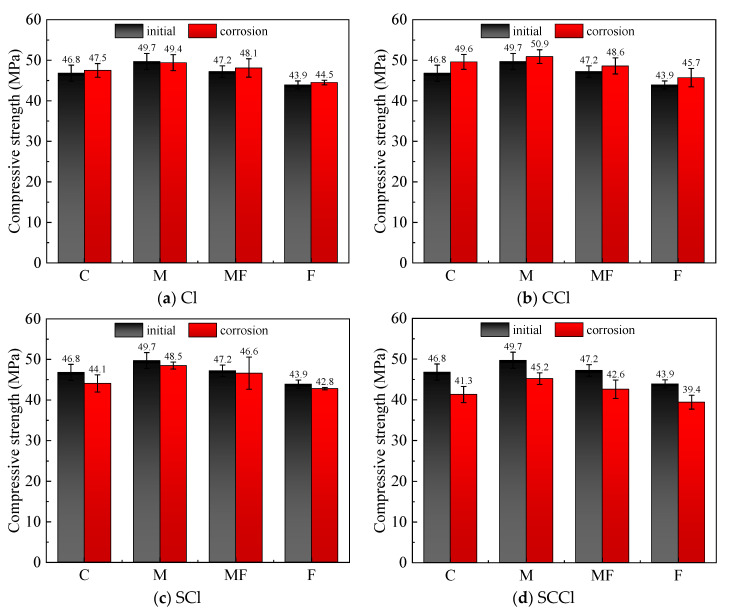
Variation in mortar strength under different erosion environments.

**Figure 7 materials-16-00007-f007:**
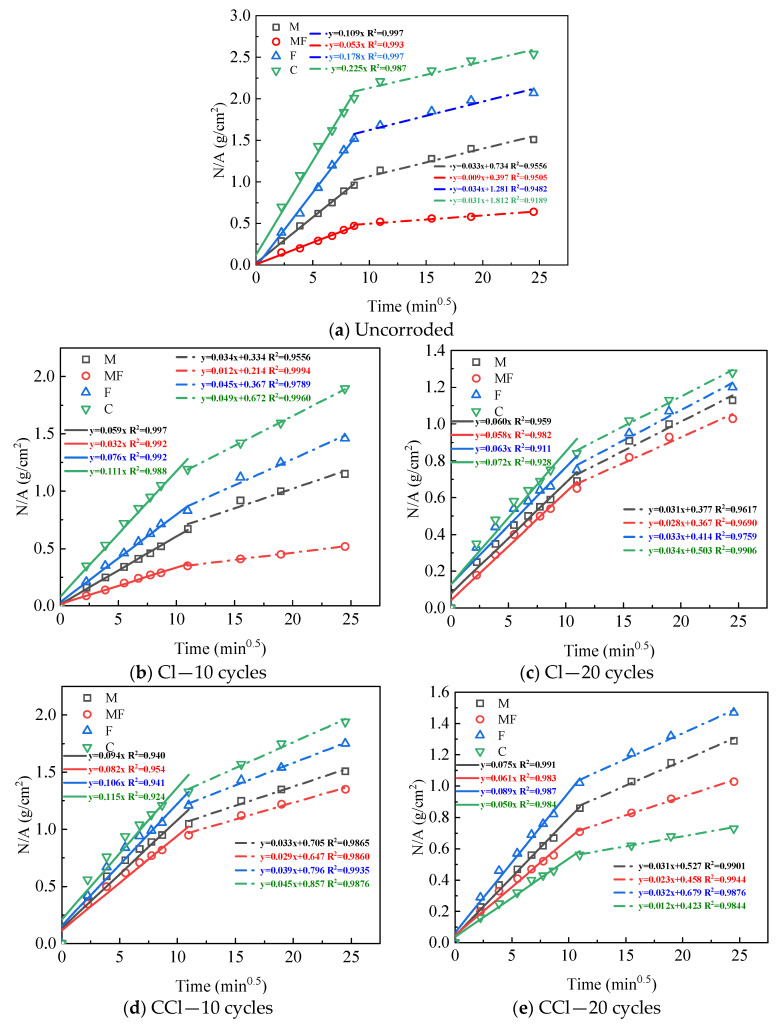
Capillary water absorption energy changes of mortar under different erosion environments.

**Figure 8 materials-16-00007-f008:**
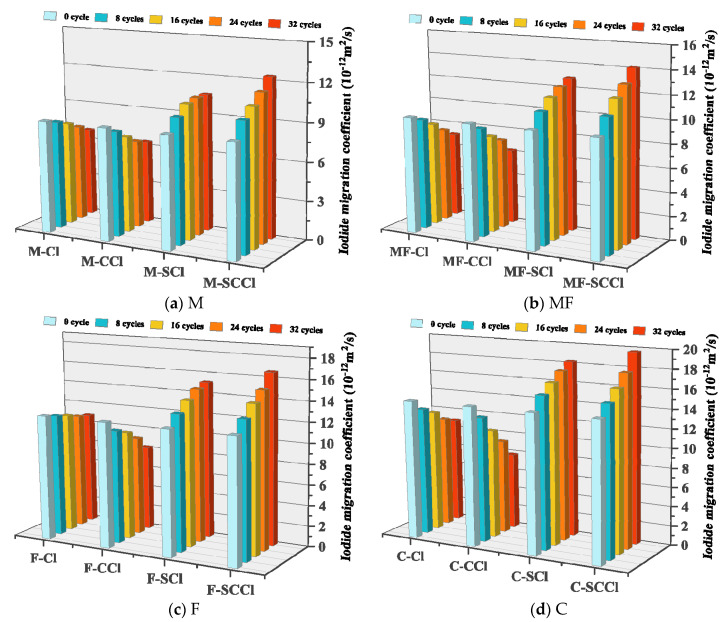
Internal migration properties of iodide ions in concrete after different environmental erosion.

**Figure 9 materials-16-00007-f009:**
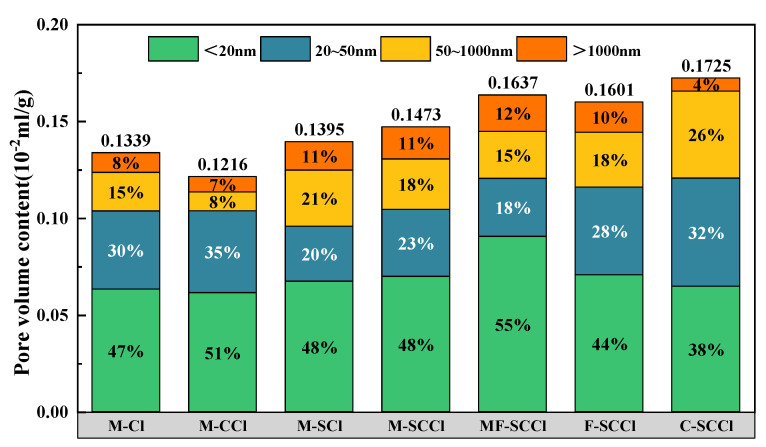
Influence of different erosion environments on the pore structure of specimens.

**Figure 10 materials-16-00007-f010:**
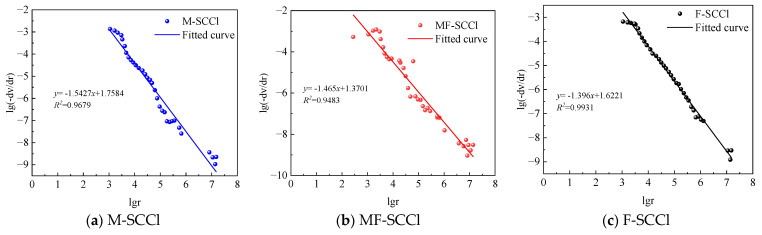
Fitting diagram for pore volume fractal dimension calculation.

**Figure 11 materials-16-00007-f011:**
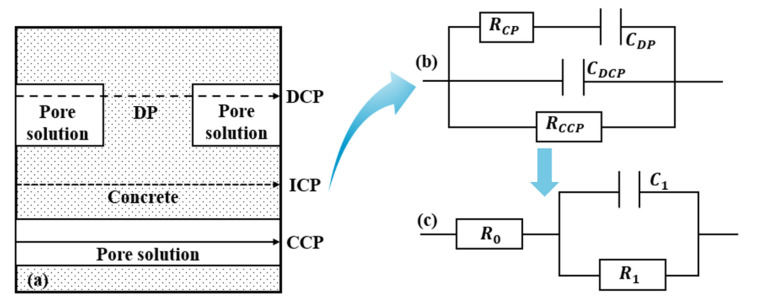
Equivalent circuit plot of concrete microstructure. (**a**) simplified model of concrete microstructure (**b**) equivalent circuit diagram (**c**) simplified equivalent circuit diagram.

**Figure 12 materials-16-00007-f012:**
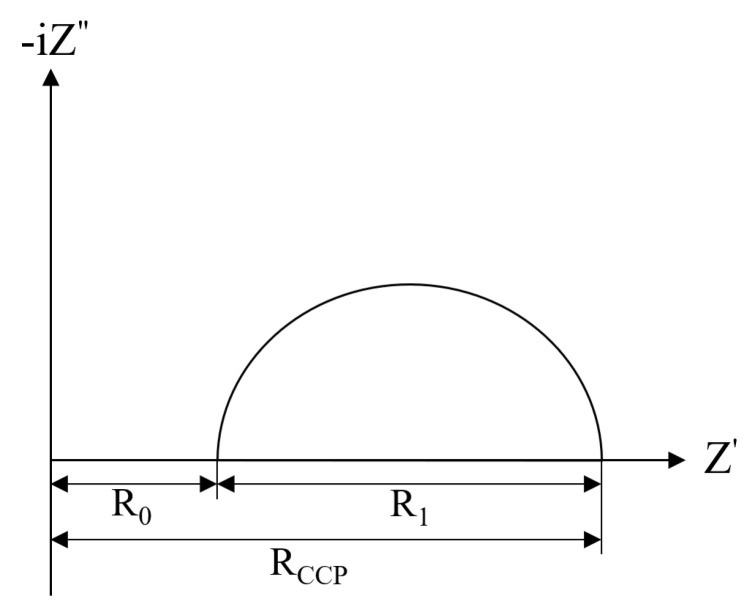
Typical Nyquist plot for concrete.

**Figure 13 materials-16-00007-f013:**
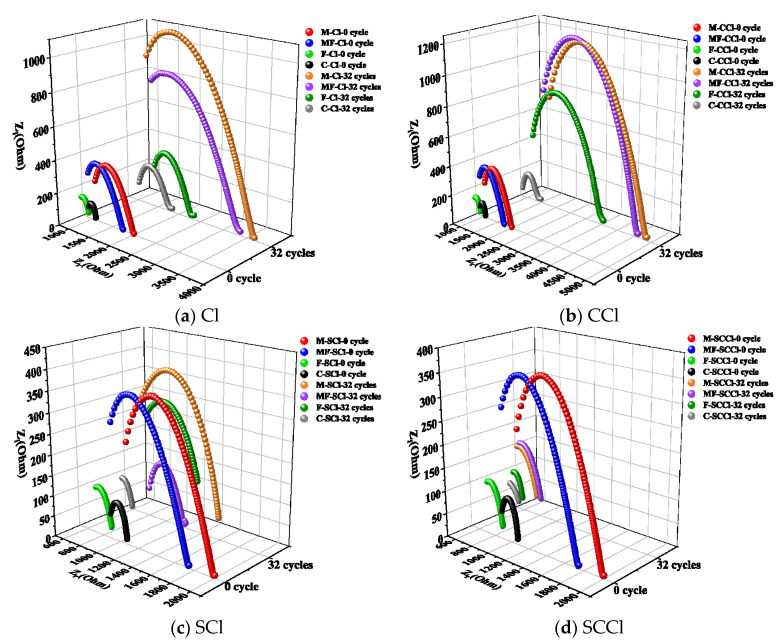
Nyquist plots of the specimens under different erosion environments.

**Table 1 materials-16-00007-t001:** Factors affecting the chloride penetration of OPC concrete.

Factors		Results	Refs.
Internal factors	Adsorption of cementitious materials	C-S-H gel has an adsorption effect on chloride ions.	[6]
	w/b	The chloride ion diffusion coefficient increases with an increase in the w/b.	[7]
	Pore structure	With the increase in hole connectivity and aperture, impermeability worsens.	[8]
	C_3_A content	There is higher impermeability with an increase in C3A content.	[9]
External factors	Corrosion time	The apparent diffusion coefficient decreases with an increase in corrosion time.	[7]
	Ambient temperature	The chloride ion diffusion coefficient increases with an increase in temperature.	[6]
	Stress	Under a stretching condition, the diffusion coefficient increases.	[6]
	Cracking	Crack zone promotes chloride ion diffusion.	[6]

**Table 2 materials-16-00007-t002:** Concrete mix ratio.

Number	Cement	Metakaolin	FLY ASH	Fine Aggregate	Coarse Aggregate	Water	Water Reducing Agent
C	450	—	—	1250	2840	176	9
M	315	135	—	1250	2840	176	9
MF	315	67.5	67.5	1250	2840	176	9
F	315	—	135	1250	2840	176	9

**Table 3 materials-16-00007-t003:** Capillary water absorption coefficient (g/(cm^2^∙min^0.5^)).

	**M-Cl**	**M-CCl**	**M-SCl**	**M-SCCl**	**MF-Cl**	**MF-CCl**	**MF-SCl**	**MF-SCCl**
0 cycle	0.109	0.109	0.109	0.109	0.053	0.053	0.053	0.053
10 cycles	0.059	0.094	0.034	0.061	0.032	0.082	0.065	0.075
20 cycles	0.060	0.075	0.029	0.026	0.058	0.061	0.019	0.018
	**F-Cl**	**F-CCl**	**F-SCl**	**F-SCCl**	**C-Cl**	**C-CCl**	**C-SCl**	**C-SCCl**
0 cycle	0.178	0.178	0.178	0.178	0.225	0.225	0.225	0.225
10 cycles	0.076	0.106	0.115	0.091	0.111	0.115	0.151	0.122
20 cycles	0.063	0.089	0.056	0.055	0.072	0.050	0.067	0.068

**Table 4 materials-16-00007-t004:** AC impedance parameters (Ω).

No.	M			MF			F			C		
Impedance Parameter	R_0_	R_1_	R_CCP_	R_0_	R_1_	R_CCP_	R_0_	R_1_	R_CCP_	R_0_	R_1_	R_CCP_
0 cycle	987	822	1809	713	884	1597	586	319	905	854	221	1075
Cl—32 cycles	1186	6466	7652	963	4181	5147	1147	3725	4872	880	583	1463
CCl—32 cycles	1161	4358	5519	1136	4169	5305	1829	2802	4631	963	856	1819
SCl—32 cycles	739	3291	4030	784	379	1163	521	913	1434	342	248	590
SCCl—32 cycles	834	1197	2031	236	2740	2976	723	835	1558	808	448	1256

## Data Availability

Not applicable.

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
