# Peer review of "Effect of Metakaolin on the Microstructural and Chloride Ion Transport Properties of Concrete in Ocean Wave Splashing Zones"

_materials, 2022, doi:10.3390/ma16010007_

Round 1

Reviewer 1 Report

Article ID: materials-2048879

This article studies the durability of marine concrete modified with metakaolin under a wave splash zone. This article presents significant merit for publication. I recommend its minor revision before acceptance.

1*The abstract presents too many results. Should be concise and summarise all parts of the paper. Include information about the addition of metakaolin.

2*Keywords: remove metakaolin and terms that are already included in the title.

3*Introduction: The first paragraph should be revised by reducing the lengths of sentences.

4 *Table 1 can be presented in a chart to make it more interesting.

5 *The rationale for the selection of metakaolin and fly ash percentages is missing. Cite relevant literature or standard.

6 *Fig. 4 the specimen size does not cubic (40 x 40 x 40mm) it looks like a prismatic sample.

7 *The compressive strength values must be presented in a table with standard deviation and coefficient of variance values.

8*The results should be correlated with literature and practical implications.

9*Fig. 10 and Fig. 11 must be refined in quality.

1*The conclusion section must be revised. Short bullet points are recommended.

1*Avoid abbreviations in the conclusion section.

1*Briefly include the significance of impedance results in the relevant section.

Reviewer 2 Report

The paper "Effect of metakaolin on microstructure and chloride ion transport properties of concrete in ocean wave splashing zone" presents a relevant theme and within the scope of this journal, and can be considered after some corrections suggested below:

(a) The abstract is generally well written, however in terms of content it is generic, i.e., the authors lack an in-depth study of the quantitative results of this research;

(b) Scientific innovation is limited in the introduction of the paper, the authors must go deeper and detail what this research differs from countless others that exist on this topic, this must be evidenced together with the objectives at the end of the introduction;

(c) The state of the art of the evaluated topic needs to be improved by the authors, note that some topics are absent and need to be known with current research, such as: 10.1016/j.cscm.2022.e01168; 10.1016/j.cscm.2022.e01034; 10.1016/j.cscm.2021.e00661

(d) Some comparative tables of results and other studies can be inserted in the introduction of the paper;

(e) At the end of the introduction, in addition to inserting the research objectives, the authors must dedicate themselves to explaining the real innovation and motivation of the study in the face of the literature in the area;

(f) Authors should better justify the criterion for choosing the cement, in addition to showing the dosage method used;

(g) The mechanical strength results must be better detailed, a statistical analysis is necessary;

(h) “The conversion relationship of the parameters in the two equivalent circuits can be obtained from the electrical theory as follows, where the parameter R0 reflects the characteristics of all continuous and discontinuous pores in concrete, and is inversely proportional to the porosity of concrete. R1 is the kinetic parameter of concrete for hydration reaction or other chemical reactions, and is a sign of the degree of hydration of concrete” This text should be better explained.

(i) The conclusion can be more objective to the readers, encompassing the main aspects and findings of the experimental program.

Round 2

Reviewer 2 Report

Ok